# Efficient Tactile Simulation with Differentiability for Robotic Manipulation

**Jie Xu**[1], **Sangwoon Kim**[1], **Tao Chen**[1], **Alberto Rodriguez**[1], **Pulkit Agrawal**[1],
**Wojciech Matusik**[1], and **Shinjiro Sueda**[2]

[1]Massachusetts Institute of Technology   [2]Texas A&M University

**Abstract:** Efficient simulation of tactile sensors can unlock new opportunities for learning tactile-based manipulation policies in simulation and then transferring the learned policy to real systems, but fast and reliable simulators for dense tactile normal and shear force fields are still under-explored. We present a novel approach for efficiently simulating both the normal and shear tactile force field covering the entire contact surface with an arbitrary tactile sensor spatial layout. Our simulator also provides analytical gradients of the tactile forces to accelerate policy learning. We conduct extensive simulation experiments to showcase our approach and demonstrate successful zero-shot sim-to-real transfer for a high-precision peg-insertion task with high-resolution vision-based GelSlim tactile sensors. The videos and code are available at: http://tactilesim.csail.mit.edu.

**Keywords:** Tactile Simulation, Tactile Manipulation, Differentiable Simulation, Sim-to-Real

## 1   Introduction

Just as humans heavily rely on rich and precise tactile cues for dexterous grasping and in-hand manipulation tasks, robots can also utilize tactile cues as an important source of sensing for interacting with the surrounding environments, especially when the visual information is unavailable or occluded. With the recent development of various tactile sensors capable of generating dense normal or shear load information [1, 2, 3, 4], researchers have been exploring how to leverage this important mode of information for robotic manipulation tasks. With the dense tactile *normal* load field, the static spatial relation between the object and the robot manipulators can easily be inferred, which is useful for tasks such as edge following [5], pose estimation [6], object reconstruction and recognition [7, 8]. On the other hand, the dense tactile *shear* force feedback more readily gives rich information about the dynamic tangential motions between the object and the manipulators, and thus can be utilized in tasks such as stable grasp [9], precise insertion [10, 11], and slip detection [12, 13, 14, 15]. However, most of the tactile manipulation work still requires a significant amount of human effort on real hardware systems for collecting data, cleverly building automatic resetting mechanisms, and carefully designing the learning strategy [10]. Such manual work can be time-consuming, cost expensive, and more importantly unsafe during policy exploration.

Due to its capability to replicate the real world with high fidelity and low cost, physics-based simulation has become a powerful recipe for learning robotic control policies [16, 17, 18, 19]. Previous work has demonstrated that the policy can be efficiently learned in simulation and successfully transferred to real robots via proper sim-to-real techniques [20, 21, 22, 23]. Despite the prevalence of simulation and the importance of tactile sensory in robotics, physics-based simulation is still under-explored to efficiently simulate dense tactile normal and shear force fields for robotic applications. Most popular simulators [18, 19] only support force-torque sensors which are attached to each robot link, only producing the contact force values at a few points on each body. Although one can acquire a dense tactile force field by attaching many small cuboids to the robot body and querying the force sensor on each small cuboid from simulation [24, 25], the obtained tactile force values are usually sparse and are unable to match the uniform force distribution on a real elastic tactile sensor such as GelSlim [1]. While researchers have also tried simulating realistic tactile feedback via pure ge-

ometric methods [5, 26], such methods typically only compute the normal tactile force and cannot simulate the tactile effects in shear directions. On the other hand, the tactile shear forces have been successfully simulated via the finite element method (FEM) [27, 28, 29] or data-driven approach [30], but these simulators suffer from expensive computation costs, and cannot be easily used for data-hungry policy-learning approaches such as reinforcement learning (RL).

We present a novel tactile simulator that can efficiently and reliably simulate both normal and shear tactile force fields covering the entire contact surface. We build upon rigid body dynamics formulation and develop a fast penalty-based tactile model which can run at 1000 frames/s for a ball-rolling experiment with a high-resolution tactile field on a single core of Intel i7-9700 CPU. Our tactile model can reasonably approximate the soft contact nature of soft tactile sensor material such as the elastomer used in GelSlim [31], generate dense tactile force fields (*e.g.,* the dense marker array on GelSlim), and is compatible with arbitrary tactile sensor spatial layout (*i.e.,* flat plane, hemisphere, etc.). Furthermore, our compact tactile formulation is differentiable, which allows the simulator to provide fast analytical gradients for the entire dynamics chain. We conduct extensive experiments in simulation to demonstrate the capabilities of our tactile simulator, including policy learning with reinforcement learning algorithms and gradient-based algorithms. We also conduct a zero-shot sim-to-real experiment for a high-precision tactile-based peg-insertion task, demonstrating that our simulator provides realistic tactile simulation.

## 2   Related Work

While there have been many physics-based simulators developed to simulate various types of robots, efficiently and reliably simulating dense tactile sensing fields is less explored. As mentioned above, most robotics simulators such as MuJoCo [18] and PyBullet [19] only support force-torque sensors that are attached to each robot link. While it is possible to augment these simulators with high-resolution tactile forces, they become computationally cumbersome. In order to acquire more realistic and dense tactile forces, Narang et al. [27, 28] and Ding et al. [29] use soft materials to model the tactile sensors and apply FEM to simulate the deformation and force fields of the tactile sensors. Despite the high fidelity of the simulated tactile feedback, these simulators suffer from expensive computation cost and are primarily used to collect supervised tactile dataset instead of learning policies which are typically data-hungry and requires fast simulations. Vision-based tactile sensors produce high-resolution tactile feedback. To simulate vision-based sensors, Wang et al. [26] and Church et al. [5] use PyBullet [19] and render the intersecting part between the object and the tactile manipulator as depth images, from which tactile information is generated. However, such purely geometry-based approaches cannot simulate the tactile effects in the shear directions such as the marker displacements of GelSlim. Si and Yuan [30] compute the marker displacement field by presenting a superposition method to approximate the FEM dynamics. While they are able to simulate the tactile shear effects, the speed of the simulation is still slow, and no control tasks are demonstrated. Bi et al. [32] build an efficient simulation specialized for a tactile-based pole swing-up task with a customized vision-based tactile sensor, but the proposed technique is not readily extensible to simulate other types of tasks and tactile sensor types. Similarly to our work, Habib et al. [33] use a spring-mass-damper model to simulate tactile normal forces, and Moisio et al. [34] use a soft bristle deflection model for simulating the tactile forces. However, there are no control tasks demonstrated to be learned with the presented simulators and no gradient information is available. In contrast to these previous works, we present a generic simulator with analytical gradients for tactile forces by leveraging the penalty-based rigid body dynamics, and we demonstrate that our simulation is efficient enough for policy learning, and simulated tactile force field can be successfully used for a sim-to-real task on the high-resolution vision-based GelSlim sensor.

## 3   Method

We now present our approach to simulate tactile forces for real-world tactile sensors. In §3.1, we introduce our flexible representation for tactile sensors. In §3.2-3.3, we present our penalty-based tactile model for simulation and derive the analytical gradients of the dynamics. In §3.4, we describe our intermediate tactile signal representation for the sim-to-real transfer of the policies.

### 3.1   Tactile Sensor Representation

Each tactile point $i$ on a sensor pad is represented by a tuple $\langle \mathbf{B}_i, \mathbf{E}_i, \xi_i \rangle$ as shown in Fig. 1. $\mathbf{B}_i$ is the rigid body the tactile point is attached to, and $\mathbf{E}_i \in \mathrm{SE}(3)$ is the position/orientation of the point in the local coordinate frame of the body, with the $\boldsymbol{x}_i$ and $\boldsymbol{y}_i$ axes in the shear-direction plane and the $\boldsymbol{z}_i$ axis along the normal tactile direction. (These axes are the same for all points for a *planar* sensor pad.) Finally, $\xi_i$ are the simulation parameters of the penalty-based tactile model, which will be introduced later in §3.2. Our representation of tactile points is flexible, allowing us to specify any number of points in arbitrary geometry layouts on a robot, and each tactile sensor can have its individual configuration parameters.

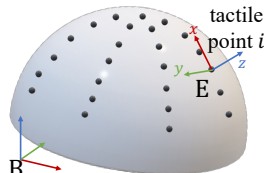

**Figure 1: Tactile Sensor Representation.**

## 3.2 Penalty-based Tactile Model

We use a penalty-based tactile model to characterize the force on each tactile point. For each point $\langle \mathbf{B}_i, \mathbf{E}_i, \xi_i \rangle$, we use the following contact model [35] to obtain the contact force at the tactile point's location represented in the local coordinate frame $\mathbf{B}_i$. (For brevity, we drop the subscript $i$.)

$$\boldsymbol{f}_n = (-k_n + k_d \dot{d})d\boldsymbol{n}, \qquad \boldsymbol{f}_t = -\frac{\boldsymbol{v}_t}{\|\boldsymbol{v}_t\|} \min(k_t \|\boldsymbol{v}_t\|, \mu \|\boldsymbol{f}_n\|), \tag{1}$$

where $\boldsymbol{f}_n$ is the contact force at the tactile point along the contact normal direction $\boldsymbol{n}$, and $\boldsymbol{f}_t$ is the contact friction force in the plane tangential to the contact normal direction. The scalar $d$ (nonpositive) is the penetration depth between the point and the collision object, and $\dot{d}$ is its time derivative. The vector $\boldsymbol{v}_t$ is the relative velocity at the contact point along the contact tangential direction. Scalars $k_n, k_d, k_t, \mu$ are contact stiffness, contact damping coefficient, friction stiffness, and coefficient of friction respectively, and they together form the simulation parameters $\xi$ of the tactile point: *i.e.,* for the $i^{th}$ tactile point, $\xi_i = \{k_n^i, k_d^i, k_t^i, \mu^i\}$. After the frictional contact force is computed for each point as $\boldsymbol{f} = \boldsymbol{f}_n + \boldsymbol{f}_t$, we transform this force into the local coordinate frame of the tactile point to acquire the desired *shear* and *normal* tactile force magnitudes:

$$T_{sx} = \boldsymbol{f}^\top \boldsymbol{x}, \qquad T_{sy} = \boldsymbol{f}^\top \boldsymbol{y}, \qquad T_n = \boldsymbol{f}^\top \boldsymbol{z}, \tag{2}$$

where $\boldsymbol{x}, \boldsymbol{y}, \boldsymbol{z}$ are the axes of frame $\mathbf{E}$.

Our penalty-based tactile model can be integrated into any simulator as long as the required values, such as the world-frame location of the tactile points, the contact normal, the collision penetration depth and its time derivatives, can be acquired from the simulator. We implement our tactile model in C++ and integrate it into differentiable RedMax (DiffRedMax) [35, 36] since DiffRedMax is open-source and readily provides all the required information for our computation, and more importantly, its differentiability allows us to make our tactile simulation differentiable with a moderate amount of modifications to its backward gradients computation.

## 3.3 Differentiable Tactile Simulation

Since we use an implicit time integration scheme for forward dynamics, the core step of gradient computation is to differentiate through the nonlinear equations of motion. We start by formulating a finite-horizon tactile-based policy optimization problem:

$$\underset{\boldsymbol{\theta}}{\text{minimize}} \ \ \mathcal{L} = \sum_{t=1}^{H} \mathcal{L}_t(\boldsymbol{u}_t, \boldsymbol{q}_t, \mathbf{v}_t(\boldsymbol{q}_t)) \tag{3a}$$

$$\text{s.t.} \ \ g(\boldsymbol{q}_{t-1}, \dot{\boldsymbol{q}}_{t-1}, \boldsymbol{u}_t, \boldsymbol{q}_t) = 0 \qquad \text{(Equations of Motion)} \tag{3b}$$

$$\boldsymbol{u}_t = \pi_\theta(\tilde{\boldsymbol{q}}_{t-1}, \tilde{\mathbf{v}}_{t-1}(\boldsymbol{q}_{t-1}), T_{t-1}(\boldsymbol{q}_{t-1}, \dot{\boldsymbol{q}}_{t-1})). \qquad \text{(Policy Execution)} \tag{3c}$$

Here, $H$ is the task horizon, $\mathcal{L}_t$ is a step-wise task-dependent reward function, $\boldsymbol{u}$ is the action (*e.g.,* joint torque), $\boldsymbol{q}$ is the simulation state (*i.e.,* joint angles), and $\mathbf{v}$ is the derived auxiliary simulation variables (*e.g.,* fingertip positions) which themselves are a function of $\boldsymbol{q}$. Eq. 3b describes the nonlinear equations of motion (§A.1). Eq. 3c represents the inference of the control policy $\pi_\theta$ to obtain the desired action given the partial observation of the simulation state $\tilde{\boldsymbol{q}}$, partial observation of the simulation computed variables $\tilde{\mathbf{v}}$, and the tactile force values $T$ from Eq. 2.

We compute the gradients $d\mathcal{L}/d\theta = (\partial \mathcal{L}/\partial \boldsymbol{u}_t)(\partial \boldsymbol{u}_t/\partial \theta)$ for policy optimization. We embed our simulator as a differentiable layer into the PyTorch computation graph and use reverse mode differentiation to backward differentiate through dynamics time integration. The first gradient, $\partial \mathcal{L}/\partial \boldsymbol{u}_t$,

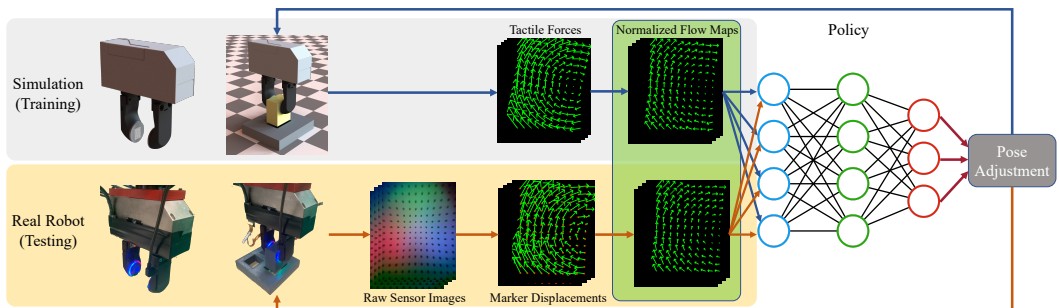

**Figure 2: Sim-to-Real Pipeline for Insertion Task (§4.5).** *Gray Box:* During training, we convert the tactile force output from the simulator into the normalized flow map representation (shaded in green). *Yellow Box:* When executing the policy on a real system, we convert the sensor output into the same normalized flow map. This intermediate representation is then treated as the observation input to a neural network policy to output the pose adjustment for the next attempt. Here we only visualize the tactile output from one tactile sensor pad.

which includes the simulation dynamics and tactile derivatives, is derived analytically, as shown below. The second gradient, $\partial\boldsymbol{u}_t/\partial\theta$, is computed by PyTorch's auto-differentiation.

We compute the gradients in reverse order. At each time step $t$, we derive $\partial\mathcal{L}/\partial\boldsymbol{u}_t$ given the analytically computed gradients with respect to the system states ($\partial\mathcal{L}/\partial\boldsymbol{q}_t$), auxiliary variables ($\partial\mathcal{L}/\partial\mathbf{v}_t$), and tactile forces ($\partial\mathcal{L}/\partial T_t$) (we drop the terms related to the time integrator here for brevity; see the full derivation in §A.2-A.3)):

$$\frac{\partial\mathcal{L}}{\partial\boldsymbol{u}_t} = \underbrace{\frac{\partial\mathcal{L}_t}{\partial\boldsymbol{u}_t}}_{a} + \underbrace{\left(\frac{\partial\mathcal{L}}{\partial\boldsymbol{q}_t} + \frac{\partial\mathcal{L}}{\partial\mathbf{v}_t}\frac{\partial\mathbf{v}_t}{\partial\boldsymbol{q}_t} + \frac{\partial\mathcal{L}}{\partial T_t}\Big(\frac{\partial T_t}{\partial\boldsymbol{q}_t} + \frac{\partial T_t}{\partial\dot{\boldsymbol{q}}_t}\frac{\partial\dot{\boldsymbol{q}}_t}{\partial\boldsymbol{q}_t}\Big)\right)}_{b} \underbrace{\frac{\partial\boldsymbol{q}_t}{\partial\boldsymbol{u}_t}}_{-A^{-1}D} . \qquad (4)$$

The right-most derivative can be computed by applying the implicit function theorem on Eq. 3b, which gives us $\partial\boldsymbol{q}_t/\partial\boldsymbol{u}_t = -(\partial g/\partial\boldsymbol{q}_t)^{-1}(\partial g/\partial\boldsymbol{u}_t)$. Writing this as $\partial\boldsymbol{q}_t/\partial\boldsymbol{u}_t = -A^{-1}D$ and combining with Eq. 4, we first solve the linear system $A^\top \boldsymbol{c} = \boldsymbol{b}^\top$ for $\boldsymbol{c}$, and then we compute the final gradient as $\partial\mathcal{L}/\partial\boldsymbol{u}_t = \boldsymbol{a} - \boldsymbol{c}^\top D$ using the adjoint approach (more details in §A.3).

### 3.4 Normalized Tactile Flow Map for Sim-to-Real

We use the GelSlim 3.0 sensor [4], which utilizes small markers to track motions in the shear direction, to demonstrate the sim-to-real capability (§4.5). There is an unavoidable sim-to-real gap between our simulator that emulates tactile forces and the physical GelSlim sensor that relies on imaging. In this section, we demonstrate how we overcome this gap by constructing a common intermediate tactile representation for policy input observation. We assume that the stiffness of the sensor along different shear directions is isotropic and that there exists a linear relationship between the displacement and the contact *shear* forces ($T_{sx}$ and $T_{sy}$ from Eq. 2) at each tactile point. We connect these two different sensor output formats via a unitless normalized tactile flow map representation.

Specifically, we use the raw tactile sensor images from the past $k$ steps from the $n$ tactile sensor pads on a real robot as our policy observation $T_{\text{image}}^{\{1:k,1:n\}}$. As shown in Fig. 2, we first detect and identify the marker positions in each image, and obtain the marker displacement field $T_{\text{displacement}}^{\{1:k,1:n\}} \in \mathbb{R}^{r\times c\times 2}$ by subtracting the marker positions in the rest configuration from their positions in the deformed configuration, where $r$ and $c$ are the rows and columns of the tactile marker array on each sensor pad, with each marker giving us the $x$ and $y$ displacement information. Then we normalize the displacement field so that the maximal length of the marker displacement across all tactile points and images is of unit length; *i.e.,*

$$T_{\text{normalized}}^{\{1:k,1:n\}} = \frac{T_{\text{displacement}}^{\{1:k,1:n\}}}{\max\big(\max_{k,n,r,c}(\|T_{\text{displacement}}^{\{k,n\}}(r,c)\|), \epsilon\big)}, \qquad (5)$$

where $\epsilon$ ensures that the output is zero when there is no any displacement on the markers (*i.e.,* no contact). We assemble these flow maps into a single tensor $T_{\text{normalized}} \in \mathbb{R}^{k\times n\times r\times c\times 2}$, which is

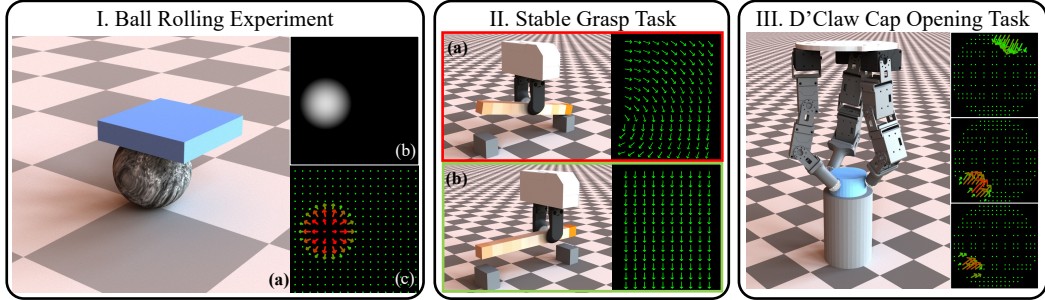

**Figure 3: (I) Ball rolling experiment:** The tactile sensors are installed on the lower surface of the pad. The depth map of the tactile normal forces is shown in (b). The tactile force field is shown in (c) with the arrow denoting the shear forces and the color denoting the magnitude of the normal force. **(II) Stable Grasp Task:** The bar is composed of 11 blocks with random densities (the deeper the color, the heavier the block). (a) An unsuccessful grasp results in rotational patterns in the tactile force field and (b) a successful grasp requires the gripper to adjust the grasp location to the center of mass of the bar. **(III) D'Claw Cap Opening Task:** The tactile sensors (white dots) are installed at the three hemisphere fingertips of the hand. We map each tactile point at one fingertip onto a 2D image plane and visualize the tactile forces field of three fingertips on the right.

our normalized tactile flow map representation. For the tactile shear forces $\{T_{sx}^{\{1:k,1:n\}}, T_{sy}^{\{1:k,1:n\}}\}$ acquired from the simulation, we conduct the same normalization process as in Eq. 5.

Intuitively speaking, the normalized tactile flow map provides directional information about the relative motion of the markers induced by the contact forces, and it also keeps the relative tactile load magnitude relationships among different sensors and different time steps so as to preserve meaningful spatial and temporal information about the contact. For our sim-to-real experiments, we only use the shear directional information from the sensor, but the same technique can also be applied to the normal directional information by normalizing the depth map of the contact surface reconstructed from the GelSlim image [4] across different frames and different sensor pads.

## 4 Experiments

We conduct extensive tactile-based experiments to demonstrate the capability of our approach.[1] We investigate the following questions: (§4.1, §4.2) Can our simulator reliably simulate the high-resolution tactile force field at a high speed for RL algorithms? (§4.3) Does the differentiability of our simulator provide advantages in policy learning? (§4.4) Is our tactile sensor representation flexible enough for sensors with arbitrary geometrical layouts? (§4.5) How does our simulated tactile force field compare to the tactile feedback from real sensors, and does our normalized tactile flow map representation help to transfer the policies learned in the simulator to a real robot?

### 4.1 Speed and Reliability: High-Resolution Tactile Ball Rolling Experiment

We design a ball rolling experiment to show the efficacy of the tactile force field generated by our simulator and to test the simulation speed. The simulation setup is shown in Fig. 3(I). A high-resolution tactile pad ($200 \times 200$ markers) touches the marble ball and moves it around. The simulation step size $h = 5$ ms, and we compute the tactile force field every 5 steps (*i.e.,* 40 Hz). Fig. 3(I) also shows the normal tactile force (represented by a depth map) and the tactile shear forces acquired from our simulator. For this example, our simulation runs at 1050 frames per second (FPS) on a single core of Intel Core i7-9700K CPU. The simulation speed can be further accelerated by simply parallelizing it across multiple CPU cores, as we do in the RL experiments.

### 4.2 RL Training: Tactile-Based Stable Grasp Task

Our tactile simulator provides the shear force information on the contact surfaces, which is critical for many manipulation tasks. Inspired by the setup in [9], we show the usage of shear force information for control and the effectiveness of our tactile simulator in a parallel-jaw bar grasping task.

---

[1]See also the supplementary video.

As shown in Figure 3(II), the task requires a WSG-50 parallel-jaw gripper to stably grasp a bar with *unknown mass distribution* in fewer than 10 attempts. The gripper has two tactile sensors with a tactile marker resolution of $13 \times 10$. The bar is composed of 11 blocks where the density of each block is randomized. The total mass of the bar is in the range $[51, 120]$ g. We consider a grasp to be a failure if the bar tilts more than $0.02$ rad after the gripper grasps a bar.

We use RL to train a control policy that determines the grasp location. The initial grasp location is the geometric center of the bar. Based on the tactile sensor readings (the only observation input to the policy), the policy outputs a delta change in the grasping location. The policy is a shallow CNN (*Conv-ReLU-MaxPool-Conv-ReLU-FC-FC*) that takes as input the two tactile sensor readings. We train the policies with PPO [37] using 32 parallel environments with 20K environment steps in total. We train the policies with 3 different random seeds and test them 320 times. The success rate is $98.5 \pm 1.8\%$. The average number of attempts taken to stably grasp the bars is 2.1, meaning the policy can compute the correct grasping location after a single failed attempt in most cases.

### 4.3 Differentiability: Tactile-Based Box Pushing Task

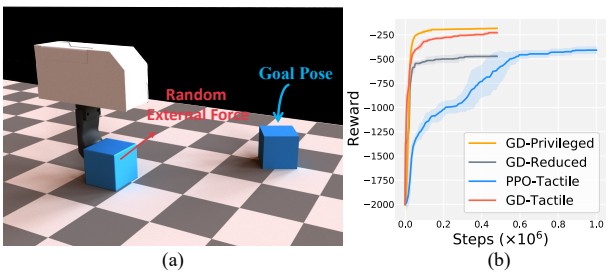

|  | POS. ERROR | ORI. ERROR |
|---|---|---|
| *GD-Privileged* | $0.037 \pm 0.002m$ | $0.043 \pm 0.003°$ |
| *GD-Reduced* | $0.126 \pm 0.009m$ | $0.255 \pm 0.021°$ |
| *PPO-Tactile* | $0.123 \pm 0.034m$ | $0.241 \pm 0.123°$ |
| *GD-Tactile* | $\mathbf{0.058 \pm 0.003m}$ | $\mathbf{0.074 \pm 0.020°}$ |

**Table 1: Metrics comparison on box pushing task.** We compute the final position/orientation errors of the best policy in each run and average the metrics from five runs for each policy variation. *GD-Privileged* gives a reference of the best possible metrics, and without the privileged state information of the box, our *GD-Tactile* achieves much better position error and rotation error than the other two variations.

**Figure 4: Tactile-based box pushing task. (a)** The goal of the gripper policy is to use its tactile feedback to push a box to a randomized target position/orientation. A time-varying external force is randomly applied to the box during the task. **(b)** the training curve for each policy variation is averaged from the five independent runs with different random seeds.

In this experiment, we design a box pushing task similar to [5] to demonstrate how we can leverage the provided analytical gradients to help learn tactile-based control policies better and faster.

**Task Specification**  As shown in Fig. 4(a), the task here is to use the same WSG-50 parallel jaw gripper as §4.2 (with only one finger kept) to push the box to a randomly sampled goal location and orientation. The initial position of the box is randomly disturbed. A random external force is applied continually on the box, which changes every $0.25$ s. More details of the task are in §B.3.

**Comparing Policy Learning Algorithms**  We train the control policies through four different combinations of learning algorithms and observation spaces. *GD-Privileged*: This variation uses the gradient-based optimizer Adam by utilizing the analytical policy gradients computed from our differentiable simulation. The policy observation contains all the privileged state information of the gripper, the box, and the goal. This policy provides an upper-bound performance reference. *GD-Reduced*: Similar to *GD-Privileged*, except that the observation space only contains the state information that can be acquired on a real system such as the gripper state and the goal. *GD-Tactile*: Other than the state information used in *GD-Reduced*, we also include the tactile sensor readings in the policy input. The policy is trained using the analytical policy gradients. *PPO-Tactile*: Similar to *GD-Tactile*, but the policy is trained by PPO.

All the policies are trained to maximize the same reward function (§B.3). We run each variation five times with different random seeds, and plot their training curves averaged from five runs in Fig. 4(b). We also randomly sample 300 goal poses and measure the final position and orientation errors between the box and the goal pose of the best policy from each run, and report the average metrics across five seeds in Table 1. The results show that when neither state information of the box nor tactile information is available (*i.e., GD-Reduced*), the policy cannot reliably push the box to the target location since the gripper has no clue when the box goes outside of the control of the gripper

due to the random initial box position perturbation and the random external forces. With tactile information feedback (*i.e., PPO-Tactile* and *GD-Tactile*), the gripper has tactile information to keep the gripper touching the box, allowing it to push the box to the goal effectively. However, the high dimensional tactile observation space results in higher computational costs with PPO, which relies on stochastic samples to estimate the policy gradients. In contrast, with the help of our differentiable tactile simulation, *GD-Tactile* makes use of the analytical policy gradients and leads to faster policy learning and better policy performance.

## 4.4 Flexibility: Tactile Sensor Simulation on Curved Surfaces

To demonstrate that our method supports tactile sensors on curved surfaces, we train a D'Claw [38] tri-finger hand to open a cap on a bottle. We put the tactile sensors on the three rounded fingertips as shown in Fig. 3(III). The sensor layout on each fingertip is a hemisphere, and we use 302 evenly-spaced tactile markers. We build a coordinate mapping to project the marker positions on the 3D surface into a $20 \times 20$ 2D array (with some empty values around the boundary). Fig. 3(III) shows that our simulation can produce reliable and realistic tactile sensor readings on a rounded fingertip.

The task is to open a cap using the D'Claw hand. The position and the radius of the cap are randomized and unknown. There is also unknown random damping between the cap and the bottle. The task is considered a success if the cap is rotated by $45°$. The only observation data that the policy gets are the angles of each joint, fingertip positions, and tactile sensor readings. This task is similar to how we open caps by just using proprioception sensory data and tactile feedback on the fingers without knowing the exact size and location of the cap. The policy outputs the delta change on the joint angles. We again use PPO to train the policy (a shallow CNN) using 32 parallel environments. To show that the tactile sensors are useful in this task, we also train a baseline policy (a simple MLP policy) where the policy only takes as input the joint angles and fingertip positions. With tactile sensor readings, the policies learn significantly faster and achieve an $87.3\%$ success rate, while the policies only achieve a $59.7\%$ success rate when tactile sensor information is unavailable. More details about task setup and results are provided in §B.4.

## 4.5 Zero-Shot Sim-to-Real: Tactile RL Insertion Task

In this experiment, we show the quality of the simulated tactile feedback when compared to the tactile sensing obtained from the real system, and demonstrate how to do zero-shot transfer for the policies learned in simulation to the real robot via our normalized tactile flow map representation.

**Task Specification**  We experiment on the tactile-RL insertion task similar to [10]. In this task, a gripper (same as in §4.2) is controlled to insert a cuboid object into a rectangle-shaped hole with a random initial pose misalignment. The insertion process is modeled as an episodic policy that iterates between open-loop insertion attempts followed by insertion pose adjustments (shown in Fig. 2). The robot has up to 15 pose correction attempts, and the robot only has access to tactile feedback from the sensors installed on both gripper fingers. For the real robot system, we use a 6-DoF ABB IRB 120 robot arm with a WSG-50 parallel jaw gripper. On each side of the gripper finger, we mount a GelSlim 3.0 tactile sensor that captures the tactile interaction between the fingers and the grasped object as a high-resolution tactile image. More details are in §B.5.

This task is more challenging than the stable grasp task (§4.2) because it not only needs to recognize the rotational pattern of the tactile field when the object contacts the front/back edges of the hole, but also needs to leverage the different magnitude relationship of two sensors' outputs to tell whether the object hits the left or the right hole edge (Fig. 5). It becomes even more challenging when the object hits the hole at four corners, because the robot must recognize nuances in the tactile pattern to decide the insertion pose adjustment. Therefore, this task requires a high-quality simulated tactile force field in order to transfer the learned policy to the real system successfully.

**Policy Learning via RL**  We train the control policies with PPO [37] for three types of misalignments. *Rotation*: The object is initialized at the hole's center and has random rotation misalignment around the vertical axis. The action of the policy is the angle adjustment in the next insertion attempt. *Translation*: The object has randomly initialized translation misalignment to the hole and no rotation misalignment. The action space in this case is two dimensional for the translational correction on the $x$-$y$ plane. *Rotation & Translation*: The object has both rotation and translation initial misalignments. The action space of the policy is three dimensional.

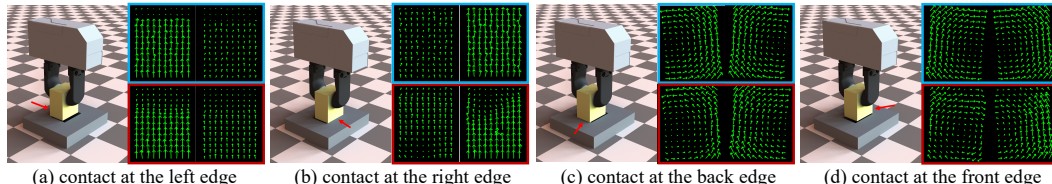

| (a) contact at the left edge | (b) contact at the right edge | (c) contact at the back edge | (d) contact at the front edge |

**Figure 5: Comparison of the normalized tactile flow maps.** The flow maps in the top blue boxes are from simulation (with noise added), while the flow maps in the bottom red boxes are produced from the real GelSlim sensor. In each box, the two flow maps (left and right) are for the two tactile pads on the two gripper fingers.

During training, we convert the simulated tactile force field into our normalized tactile flow map representation (§3.4), and treat the resulting tactile flow map as a $13 \times 10$ flow "image" with 4 channels (2 sensors and 2 shear components of tactile forces). We model the policy by a convolutional RNN to leverage more information from previous attempts. For better sim-to-real performance, we also apply the domain randomization technique [39] on contact parameters, tactile sensor parameters, grasp forces, grasp height and tactile readings, to increase the robustness of the learned policies. More details of policy learning are provided in §B.5.

**Experiment Results**   We first qualitatively compare the normalized tactile flow maps generated by simulation and by real GelSlim sensors. We plot the normalized tactile flow maps at four representative contact configurations (*i.e.,* object contacts at different edges of the hole) [10] in Fig. 5, which shows that our simulation is able to produce highly realistic patterns in those contact configurations. We then deploy the policies learned in simulation on real hardware and quantitatively test its zero-shot performance by conducting 100 insertion experiments under different initial pose misalignments. As reported in Table 2, our zero-shot policy transfer achieves 100%

| TASK | SUCCESS | ATTEMPTS |
|------|---------|----------|
| *R*    | 100%    | 1.53     |
| *T*    | 100%    | 2.33     |
| *R&T*  | 83%     | 4.81     |

**Table 2: Zero-shot sim-to-real performance of the tactile RL insertion policies.**

success rates on *Rotation* and *Translation* tasks. We also calculate the average number of pose corrections for successful experiments. The average number of pose corrections is 1.53 for the *Rotation* task and 2.33 for the *Translation* task, which means that the policy is able to successfully infer the pose misalignment after just one or two failed attempts in most experiments. Given that the policies are purely trained with simulated tactile data, the high success rates indicate that our simulation is able to produce normalized tactile flow maps with highly realistic tactile patterns and magnitudes to help the gripper to infer the exact adjustment. For challenging *Rotation & Translation* task, our zero-shot transferred policy also achieves 83% success rate and 4.81 pose corrections on average. For comparison, Dong et al. [10] achieve 89.6% success rate and 5.42 times pose adjustments for the cuboid object, with a policy trained directly on the real hardware from a pre-trained policy and with a carefully designed task curriculum. On the other hand, our policy is trained from scratch only in simulation without observing any real-world data.

## 5   Limitations and Future Work

We presented an efficient differentiable simulator that can handle dense tactile force fields with both normal and shear components. When the tactile pad is *very* soft (*e.g.,* TacTip), its dynamics cannot be well approximated by our penalty-based approach. An interesting direction to explore is how to efficiently simulate such soft tactile sensors. We demonstrated with the box pushing task (§4.3) the potential advantage of differentiable simulators. However, how to effectively leverage analytical gradients for more complex tactile-based tasks is still an open question and it may require more advanced policy learning algorithms [40]. Furthermore, in the sim-to-real experiment (§4.5), the zero-shot success rate of *Rotation & Translation* is not perfect. This is probably due to some intricacies of the real hardware that are difficult to model in our simulation. We believe that further fine-tuning the learned policies with a few shots on the real hardware will likely lead to improved performance.

**Acknowledgments**

We thank the anonymous reviewers for their helpful comments in revising the paper. We also thank Toyota Research Institute (TRI), ONR MURI (N00014-22-1-2740), and the National Science Foundation (CAREER-1846368) for providing funds to support this work.

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
