# OpenReview forum: "Efficient Tactile Simulation with Differentiability for Robotic Manipulation"
_robot-learning.org/CoRL/2022/Conference — CoRL 2022 Poster_

### Official Review · Reviewer_VkwB · 2022-07-24

**Originality:** Very Good
**Technical Quality:** Very Good
**Clarity Of Presentation:** Very Good
**Impact:** 4

**Recommendation:**

Strong Accept: I recommend accepting the paper and will argue for my recommendation even if other reviewers hold a different opinion.

**Summary:**

The paper presents a novel differentiable physics simulator that includes differentiable tactile sensing simulation, and shows that it can facilitate Sim-to-Real Reinforcement Learning on robots equipped with GelSight tactile sensors.

**Issues:**

[1] Line 133-135 "...which includes the tactile derivatives, is derived analytically, as shown below...": I cannot seem to find the reason (on this paper) WHY do you need to derive the derivatives analytically, instead of relying on PyTorch’s auto-differentiation... Can the authors please elaborate?

[2] In Equation (1) on Line 103-104, do you need to approximate the min() function with a softmin() function instead (for smoothness reasons and better differentiability)?

[3] Line 140-141: I am a little lost when trying to follow the derivation that involves c (boldface c) in these lines. Could you please try to re-phrase to improve the clarity?

[4] On Experiment 4.5, the authors show Sim-to-Real RL evaluation, but I cannot fully understand WHY it transfers quite successfully from simulation to the real hardware. Is it possible to add a more direct numerical comparisons between the data that the simulation produce as compared to the data that the hardware produce, and compare them side-by-side? So far, I only see in Figure 2 some pictorial comparisons between the data of sim vs real, but numerical comparisons would probably help in understanding better WHY the Sim2Real transfer was successful...

**Quality Of The Limitations Section:**

Limitations are addressed clearly

**Reviewer Expertise:**

5: The reviewer is absolutely certain that the evaluation is correct and very familiar with the relevant literature

**Robotics Focus:**

Sufficient demonstration on hardware

**Strengths And Weaknesses:**

Strengths:
[1] The paper presents a novel differentiable tactile simulators for Robot Learning research.
[2] Comprehensive experiments and its summary (i.e. what message does the experiment outcome represents) are presented.

Weaknesses:
[1] Please see comments and suggestions in the "Issues" section below, and please try to address them.

**Summary Of Recommendation:**

I recommend for a "Strong Accept" on this paper, given the paper's novelty and contributions, as well as comprehensive experiments and excellent presentation of the paper and results.

---

> ### Author Response · Authors · 2022-08-24
> **Response to Reviewer VkwB**
>
> We sincerely thank reviewer VkwB for the feedback on our paper. We address the reviewer’s concerns as follows:
>
> &nbsp;
>
> **Q: Why do we need to derive the gradients analytically?**
>
> **A:** There are two main reasons for deriving the gradients analytically rather than resorting to PyTorch auto-differentiation. (1) For stability purposes, our simulation uses an implicit time integration scheme where a non-linear equation is solved in each time step. Solving a non-linear equation is not supported by most current auto-differentiation engines. (2) For computational speed purposes, some gradient calculation techniques are used such as the Adjoint method to leverage the special structure of the dynamics equations. Blindly using auto-differentiation tools will not take those advantages thus slowing down the gradient computation. Because of those considerations, we follow the DiffRedMax and implement the core of the simulation in C++ and wrap it as a PyTorch layer so that we are able to easily use it as a regular PyTorch function and include it into PyTorch computational graph without sacrificing the stability and speed of the simulation.
>
> &nbsp;
>
> **Q: Do we approximate the min() function by a smoother formulation (e.g. softmin) for better differentiability?**
>
> **A:** We do not approximate the min() function in our implementation since it seems to be sufficiently good for our experiments. However, we agree with the reviewer that an approximation of min() function can probably improve the smoothness and provide better differentiability.
>
> &nbsp;
>
> **Q: Better clarity about the derivation involving $\mathbf{c}$.**
>
> **A:** We apologize for the missing clarity about $\mathbf{c}$. There is a typo in the equation in line 141, which should instead be $d\mathcal{L}/d\mathbf{u}_t = \mathbf{a} + \mathbf{c}^\top D$. We provide a bit more details in Appendix A.3. Specifically, we denote $\mathbf{a} = \frac{\partial\mathcal{L}_t}{\partial \mathbf{u}_t}, \mathbf{b} = \frac{\partial\mathcal{L}}{\partial \mathbf{q}_t} + \frac{\partial\mathcal{L}}{\partial\mathbf{\mathsf{v}}_t}\frac{\partial\mathbf{\mathsf{v}}_t}{\partial\mathbf{q}_t} +
>         \frac{\partial\mathcal{L}}{\partial T_t}\big(\frac{\partial T_t}{\partial \mathbf{q}_t}+\frac{\partial T_t}{\partial \dot{\mathbf{q}}_t}\frac{\partial \dot{\mathbf{q}}_t}{\partial \mathbf{q}_t}\big)
> , A = \frac{\partial g}{\partial \mathbf{q}_t}, D = \frac{\partial g}{\partial \mathbf{u}_t}$, where $\mathbf{a}, \mathbf{b}$ are row vectors and $A, D$ are matrices. Then Eq. (4) becomes $\frac{d \mathcal{L}_t}{d \mathbf{u}_t} = \mathbf{a} - \mathbf{b} A^{-1}D$. To compute $-\mathbf{b} A^{-1}D$, we first compute $\mathbf{c}^\top = -\mathbf{b}A^{-1}$ where $\mathbf{c}$ is a column vector. We obtain $\mathbf{c}$ by solving the $A^\top\mathbf{c} = -\mathbf{b}^\top$. Once we have $\mathbf{c}$ computed, the Eq. (4) becomes $\frac{d \mathcal{L}_t}{d \mathbf{u}_t} = \mathbf{a} + \mathbf{c}^\top D$. We will revise the paper to increase the clarity of this part.
>
> &nbsp;
>
> **Q: Why sim-to-real is so successful? Any numerical comparison?**
>
> **A:** The success of the sim-to-real comes from two parts. (1) Normalized tactile flow map. In the simulation, we simulate the tactile force field, while in real hardware, the GelSlim sensor outputs the displacement field of the marker points. We bridge these two different tactile representations by normalizing them (Eq. 5); thus the trained policy can be less sensitive to the mismatching magnitudes of tactile representations between simulation and the real world. (2) Domain randomization. We randomize the simulation parameters within a wide range to further increase the robustness of the learned policies. Besides Fig 2, we provide more direct qualitative tactile field comparisons between simulation and real sensor in Fig 5. Fig 5 includes four basic cases for the object-hole contact in the insertion task. In the four subfigures of Fig 5, the mean error of the normalized tactile flow maps between sim (blue) and real (red) are 0.1482, 0.2334, 0.1314, and 0.1958, respectively. Please note that each arrow was normalized to be in the range [-1, +1] in X and Y, and so these errors roughly amount to 6.5% to 11.5% of the allowable range. Considering the amount of noise in the hardware, we believe these numbers are quite respectable. Importantly, the **patterns** of the tactile forces match well in those representative cases, which we believe is essential for our experiments.

---

### Official Review · Reviewer_BP21 · 2022-07-31

**Originality:** Very Good
**Technical Quality:** Very Good
**Clarity Of Presentation:** Excellent
**Impact:** 4

**Recommendation:**

Weak Accept: I recommend accepting the paper, but will not argue for my recommendation if the majority of other reviewers have a different opinion.

**Summary:**

This paper presents a differentiable tactile simulator for robot manipulation. It implemented a penalty-based tactile model based on a differentiable RedMax to compute an analytical gradient for policy optimization. When performing the sim-to-real transfer, it proposes compensating the sim-to-real gap by seeing the sim and real tactile differences and normalizing the deformation.

This study performed various experiments to confirm speed, flexibility, and differentiability. The proposed simulation can run with 1050 FPS and perform various tasks and shapes of the tactile sensors. In the box pushing tasks, the proposed analytical gradient policy optimization showed faster convergence than the RL-based methods. Finally, it performed sim-to-real insertion tasks and demonstrated the zero-shot transfer.

**Issues:**

As described in the weakness, this paper needs to explain what "efficient" means in more detail. I would like to see whether gradient optimization can be used in different tasks.

**Quality Of The Limitations Section:**

Limitations are addressed clearly

**Reviewer Expertise:**

2: The reviewer is willing to defend the evaluation, but it is quite likely that the reviewer did not understand central parts of the paper

**Robotics Focus:**

Sufficient demonstration on hardware

**Strengths And Weaknesses:**

STRENGTH

This paper is excellently well written. The figures are beautiful. It made much effort to perform experiments in various environments. The experimental section is explained in detail. I believe this simulation must be helpful for users who are interested in tactile manipulation. The demo video is also beneficial to see if the simulation is close to the behaviors of the actual tactile sensors. I would highly appreciate it if this study released an open-source code for this simulation.

WEAKNESS

Although this study is excellent, it needs to address some minor points.

a) The meaning of "efficiency" sounds unclear. Which computational or sample efficiency does this study mean? If it means computational efficiency, this study needs more comparisons or discussions in section 4.1. How much faster is the proposed simulation than the existing simulators? Please provide quantitative explanations. Besides, I would recommend revising the manuscript entirely to clarify the efficiency.

b) The proposed analytical gradient policy optimization demonstrated faster convergence in the box pushing. I would like to see the performance in the other tasks, such as insertion using the gradient method. This might be very important to support the usefulness of the proposed method.

c) The soft robotics field is also exploring differentiable simulations. More discussions about differentiable models with soft robots might help enhance the novelty. The following reference might be an example.

Dubied, M., Michelis, M. Y., Spielberg, A., & Katzschmann, R. K. (2022). Sim-to-Real for Soft Robots Using Differentiable FEM: Recipes for Meshing, Damping, and Actuation. IEEE Robotics and Automation Letters, 7(2), 5015-5022.

**Summary Of Recommendation:**

Overall, this study is excellent. I would recommend weak acceptance for this paper.

---

> ### Author Response · Authors · 2022-08-24
> **Response to Reviewer BP21**
>
> We sincerely thank reviewer BP21 for the feedback on our paper. We address the reviewer’s concerns as follows:
>
> &nbsp;
>
> **Q: Code release.**
>
> **A:** We will release the full code repository of the simulation, the tactile control environments, and the policy training upon acceptance.
>
> &nbsp;
>
> **Q: The definition of “efficiency” is unclear.**
>
> **A:** We apologize for the confusion. We refer “efficiency” to computational efficiency in our work. In other words, the “efficiency” of a simulator means that it can simulate tactile signals at a fast speed. The existing simulations for simulating both tactile normal and tactile shear forces are based on FEM methods. The state-of-the-art FEM-based tactile simulator [1] takes 5.5 seconds for each quasi-static object indention experiment, and it is used to generate tactile dataset and has not been utilized for data-expensive control policy learning tasks yet which usually requires higher simulation speed. As suggested, we will revise the paper to explicitly clarify the meaning of efficiency.
>
> &nbsp;
>
> **Q: The performance of differentiable simulation in other tasks.**
>
> **A:** Thanks for bringing up this suggestion. The goal of our work is to offer a new tool to the robotics community, thus we use a most straightforward approach (i.e. gradient descent) to demonstrate the differentiability feature of our simulation. However, as we discuss in the limitation section, how to effectively leverage the analytical gradients provided by the simulator on more complex tasks is still an open problem and an active research topic in the field [2, 3, 4], and is out of the scope of this work. We do believe our developed simulation tool can offer more opportunities to the robotics learning community and facilitate the development of more advanced tactile-based policy learning algorithms utilizing the differentiable simulation.
>
> &nbsp;
>
> **Q: Discuss the comparison with differentiable soft robot simulation.**
>
> **A:** The main advantage of our rigid body-based differentiable tactile simulation over differentiable FEM simulation is the speed. For example, the work [5] mentioned by the reviewer utilizes the differentiable FEM simulation developed by [6]. [6] leverages the projective dynamics technique to accelerate the FEM simulation with a non-linear Neo-Hookean material model, and its running speed is 7 FPS for the simplest rolling sphere example in their paper. In contrast, our simulation can run at 1000 FPS for a ball rolling experiment on one single CPU core. On the other hand, the advantage of [5] over our simulator is in that [5] uses a non-linear Neo-Hookean material model which allows them to simulate high-elasticity objects/robots. We will clarify this comparison in the revised paper.
>
> &nbsp;
>
> [1] Y. Narang, B. Sundaralingam, M. Macklin, A. Mousavian, and D. Fox. Sim-to-real for robotic tactile sensing via physics-based simulation and learned latent projections. 2020.
>
> [2] C. D. Freeman, E. Frey, A. Raichuk, S. Girgin, I. Mordatch, and O. Bachem. Brax - a differentiable physics engine for large scale rigid body simulation. 2021.
>
> [3] J. Xu, V. Makoviychuk, Y. Narang, F. Ramos, W. Matusik, A. Garg, and M. Macklin. Accel- erated policy learning with parallel differentiable simulation. 2021.
>
> [4] H.J. Suh, M. Simchowitz, K. Zhang, R. Tedrake. Do Differentiable Simulators Give Better Policy Gradients? 2022.
>
> [5] M. Dubied, M. Y. Michelis, A. Spielberg, & R. K. Katzschmann. Sim-to-Real for Soft Robots Using Differentiable FEM: Recipes for Meshing, Damping, and Actuation. 2022.
>
> [6] T. Du, K. Wu, P. Ma, S. Wah, A. Spielberg, D. Rus, W. Matusik. ​​DiffPD: Differentiable Projective Dynamics. 2021.

---

### Official Review · Reviewer_4x5o · 2022-08-01

**Originality:** Very Good
**Technical Quality:** Excellent
**Clarity Of Presentation:** Excellent
**Impact:** 4

**Recommendation:**

Strong Accept: I recommend accepting the paper and will argue for my recommendation even if other reviewers hold a different opinion.

**Summary:**

This paper presents a fast differentiable approach to vision-based tactile simulation that captures both the normal and shear tactile force measurements of tactile sensors of varying geometries. Experiments both in simulation and on the real robot showed that the analytical gradients provided by the tactile simulation accelerated the learning of different contact-rich tasks such as ball rolling, stable grasp adjustment, and peg insertion task. The speed of the tactile simulation (*1000 fps*) and impressive task learning performance suggests that this can be a very impactful work.

**Issues:**

- Some comments or discussion around selection of material parameters of the tactile sensor is needed. What's a good range and why? How can the parameter be determined for a different sensor e.g. DIGIT


**Quality Of The Limitations Section:**

Limitations are addressed clearly

**Reviewer Expertise:**

5: The reviewer is absolutely certain that the evaluation is correct and very familiar with the relevant literature

**Robotics Focus:**

Sufficient demonstration on hardware

**Strengths And Weaknesses:**

Strengths
- The paper tackles an important challenge of simulating vision-based tactile measurement using a novel penalty-based model. The proposed approach not only captures the normal tactile forces as in a number of previous works, but also shear forces which has been expensive to compute until now
- Impressively fast simulation speed was reported; achieving 1000 frames per second on a single core CPU.
- Effective sim-to-real transfer of policies learned using the proposed tactile simulation method is very valuable. Given that many of the high resolution tactile sensors like GelSlim are pretty expensive, simulating them in a high fidelity way enables transferrable policy learning in simulation and can improve the useful life of the real sensors. The paper demonstrated zero-shot sim-to-real transfer of a policy learned entirely in simulation.
- The paper is very well organized and clear. The derivations and details in the supplementary material aids the full understanding of the work. The video presents the experimental results in a very clear manner; tasks execution with overlaid tactile measurements were presented for the simulated task and real robot insertion task.

Weaknesses
- It is unclear how the simulation parameters $\xi$ of the tactile point are determined for different tactile elastomer materials. This is an important detail for the reproducibility of the work either on the illustrated GelSlim sensor or for application to other tactile sensors.
- The paper did not mention if the simulator will be released. Releasing the code will significantly increase the potential impact of the work.

**Summary Of Recommendation:**

The paper proposed a strong tactile simulation approach, and presented rigorous experiments to evaluate the fidelity and performance of the proposed methods. The ability to achieve zero-shot performance is impressive; this validates the fidelity of the simulator and showcases the potential impact of the work.

---

> ### Author Response · Authors · 2022-08-24
> **Response to Reviewer 4x5o**
>
> We sincerely thank reviewer 4x5o  for the feedback on our paper. We address the reviewer’s concerns as follows:
>
> &nbsp;
>
> **Q: How are the tactile simulation parameters \xi determined?**
>
> **A:** Since we use a normalized tactile flow map as an intermediate representation to minimize the mismatch between simulation and the real world, we actually did not spend much time on obtaining those parameters. In our current work, we determine the tactile simulation parameters by just setting them as reasonable values and manually tuning them a bit based on the observed differences between the simulated tactile pattern and the tactile pattern on the real GelSlim sensor. For reproducibility, we report the parameters we used in the simulation in Table 7 in the Appendix. However, we do think having better simulation parameters will help achieve better sim-to-real performance. Since differentiable simulation can also provide the gradients of the simulation parameters, it would be an interesting future direction about how to leverage differentiable simulation to calibrate the simulation parameters easier and faster.
>
> &nbsp;
>
> **Q: Code release.**
>
> **A:** We will release the full code repository of the simulation, the tactile control environments, and the policy training upon acceptance.

---

### Official Review · Reviewer_Led2 · 2022-08-06

**Originality:** Good
**Technical Quality:** Very Good
**Clarity Of Presentation:** Very Good
**Impact:** 3

**Recommendation:**

Weak Accept: I recommend accepting the paper, but will not argue for my recommendation if the majority of other reviewers have a different opinion.

**Summary:**

The paper develops a differentiable tactile simulator that models the tactile sensor with discrete points on the surface of the sensor and calculates the normal and shear force at these points from interaction with objects. The analytical equations used to compute these forces are differentiable, thereby enabling the use of gradients from tactile contact in downstream tasks. The paper uses this interaction function as a differentiable layer in pytorch and enables differentiation through dynamics. The paper runs experiments in simulation on a few different manipulation tasks and also runs object insertion experiments in the real world to validate the approach.

**Issues:**

The main drawback of the paper is that the developed simulation is limited to only force based interaction tasks and also to sensors that have a near rigid external surface. Running a real world experiment with a soft/deformable object or using a softer tactile sensor would help in understanding the approach's limitation. A discussion on lack of contact field estimation from the simulation is needed in the limitations section. Especially discussing whether we can use the simulation for pose estimation tasks.

If the above limitations are in fact true, stating these in the abstract and changing the title to say "rigid-body" tactile simulation for manipulation would be helpful as readers could misunderstand the paper as being able to simulate all modalities from a tactile sensor.

**Quality Of The Limitations Section:**

Additional details required

**Reviewer Expertise:**

3: The reviewer is fairly confident that the evaluation is correct

**Robotics Focus:**

Sufficient demonstration on hardware

**Strengths And Weaknesses:**

The paper does a good job in validating the proposed approach, especially compared to PPO (where there is a lack of gradients from tactile interactions). The variety of simulation experiments also help the readers in understanding the importance of gradients from tactile sensors.

Some concerns I have with the paper are listed here:
1. The paper mentions that existing FEM simulation models for tactile sensing (line 66) are slow which is false. Existing FEM simulation models are sufficiently fast while also giving high fidelity simulation of the contact area. The proposed method fails to simulate the surface deformation which is very common in robotic tactile sensors. Hence, the paper could instead mention that existing FEM simulation models lack gradients, preventing their use in gradient based optimization.

2. Since the method models contact interaction with rigid body dynamics, how will this affect interaction with soft/deformable objects? Running some experiment in the real world to quantify this would be interesting.


**Summary Of Recommendation:**

1. The paper’s approach cannot accurately simulate contact deformations, thereby making the method only work for tactile sensors that have a very stiff external surface.
2. The paper’s simulation also cannot be used for pose estimation, which is a common application domain for tactile sensing as the simulation only reasons about forces at contact points.

---

> ### Author Response · Authors · 2022-08-24
> **Response to Reviewer Led2**
>
> We sincerely thank reviewer Led2 for the feedback on our paper. We address the reviewer’s concerns as follows:
>
> &nbsp;
>
> **Q: Comparison to FEM-based simulation.**
>
> **A:** There are indeed differentiable FEM simulations [1, 2]. The main advantage of our rigid body approximation over FEM simulation is the simulation speed. For example, [3] uses FEM simulation with the linear-elastic material model to simulate the tactile sensor feedback, and it takes 5.5 seconds for each quasi-static object indention experiment even with GPU parallelization. For differentiable FEM simulators, [2] leverages the projective dynamics technique to accelerate the FEM simulation with a non-linear Neo-Hookean material model, and its running speed is 7 FPS for the simplest rolling sphere example in their paper. In contrast, our simulation can run at 1000 FPS for a ball rolling experiment on one single CPU core. We will clarify this comparison in the revised paper. Furthermore, we agree that one advantage of FEM simulation is that it is able to simulate surface/interior deformation; however whether it is necessary to simulate the deformation is unclear in many robotics problems, since we are unable to obtain the deformation information in the real hardware during testing time.
>
> &nbsp;
>
> **Q: Interaction with soft/deformable objects.**
>
> **A:** Thanks for raising this point. To enable date-intensive applications such as policy learning, we trade part of the accuracy of the simulator for its speed. Our simulator is based on rigid body dynamics for its fast simulation speed while utilizing a soft penalty-based contact model to approximate some softness of the object interaction in the real world. Our simulation works reasonably well for relatively stiff elastic materials such as the GelSlim pad. However, as we have described in the limitation section, it is one major drawback of our simulator that it is unable to accurately simulate **very** soft tactile sensors/objects. To simulate **very** soft/deformable objects, the state-of-the-art simulation techniques require to use FEM with a non-linear material model, which is known to be computationally expensive [4].
>
> &nbsp;
>
> **Q: Can we use the proposed simulator for pose estimation tasks?**
>
> **A:** Our simulator can be used for pose estimation tasks without any major modifications. The simulated tactile force field is generalizable to different tasks. Take the GelSlim sensor as an example, all the tasks that can be achieved by a GelSlim sensor should be able to be reproduced in our simulator. We achieve this by treating the normalized tactile flow map as an intermediate representation between the simulated tactile force field and the tactile sensing signals in real hardware. As an example, the tactile normal force field is equivalent to the contact depth map which can be reconstructed from a vision-based tactile sensor (e.g. GelSlim) and can be used to estimate the pose of the object.
>
> &nbsp;
>
> [1] M. Geilinger, D. Hahn, J. Zender, M. Bacher, B. Thomaszewski, S. Coros. ADD: Analytically Differentiable Dynamics for Multi-Body Systems with Frictional Contact. 2020.
>
> [2] T. Du, K. Wu, P. Ma, S. Wah, A. Spielberg, D. Rus, W. Matusik. ​​DiffPD: Differentiable Projective Dynamics. 2021.
>
> [3] Y. Narang, B. Sundaralingam, M. Macklin, A. Mousavian, and D. Fox. Sim-to-real for robotic tactile sensing via physics-based simulation and learned latent projections. 2021.
>
> [4] Y. S. Narang, K. Van Wyk, A. Mousavian, and D. Fox. Interpreting and predicting tactile signals via a physics-based and data-driven framework. 2020.

---

### Author Response · Authors · 2022-08-24
**General Response to AC and Reviewers**

We thank the reviewers for their insightful feedback and thank the AC for the metareview. We are glad that the reviewers found that:

1. The tackled problem is challenging and important. (Reviewer 4x5o)

2. The developed simulation is novel and can be a valuable contribution to the robotics community. (Reviewer 4x5o, BP21, VkwB)

3. The experiments are rigorous and extensive, and the results are convincing. (Reviewer hP5o, Led2, 4x5o, BP21, VkwB)

4. The paper is well written and easy to follow. (Reviewer hP5o, Led2, 4x5o, BP21, VkwB)

5. The video is clear and helpful for understanding the paper. (Reviewer 4x5o, BP21)

A concern raised by two of the reviewers (Led2, BP21) was about comparison to FEM-based simulation. We will make the comparison clearer in the related work section in the revised version. Another common request is the code release; we will release the code upon acceptance. We have addressed the remainder of the reviewers' concerns individually.

---

### Meta-Review · Area_Chair_hP5o · 2022-08-10

**Recommendation:** Accept (Poster)
**Confidence:** 5

**Metareview:**

This paper presents a differentiable and fast contact model including shear force that enables efficient reinforcement learning with simulated tactile sensors of arbitrary shapes. The model is used with PPO to learn 3 tasks in simulation. The authors also present sim-to-real transfer in the insertion task.

The paper is well written and easy to follow, and simulation and hardware evaluations are convincing. The authors' responses successfully addressed the concerns such as comparison to FEM-based methods.

While the main contribution (analytical derivative of a contact model) is useful beyond robot learning, it is a tool that also benefits this community as demonstrated by its applications to RL in the paper.

**Best Paper Nomination:**

No